# Fungi Present in the Clones and Cultivars of European Cranberry (*Vaccinium oxycoccos*) Grown in Lithuania

**DOI:** 10.3390/plants12122360

**Published:** 2023-06-18

**Authors:** Jolanta Sinkevičienė, Aušra Sinkevičiūtė, Laima Česonienė, Remigijus Daubaras

**Affiliations:** 1Department of Agroecosystems and Soil Sciences, Agriculture Academy, Vytautas Magnus University, K. Donelaičio Str. 58, LT-44248 Kaunas, Lithuania; 2Botanical Garden, Vytautas Magnus University, Z.E. Žiliberio 6, LT-46324 Kaunas, Lithuania; 3Faculty of Odontology, Lithuanian University of Health Sciences, J.Lukšos-Daumanto 2, LT-50106 Kaunas, Lithuania

**Keywords:** cranberry, *Vaccinium oxycoccos*, clone, cultivars, pathogenic fungi

## Abstract

Fungi are associated with the European cranberry (*Vaccinium oxycoccos* L.) and play important roles in plant growth and disease control, especially in cranberry yields. This article presents the results of a study which was aimed to investigate the diversity of fungi found on different clones and cultivars of the European cranberry grown in Lithuania, causing twigs, leaf diseases and fruit rots. In this study seventeen clones and five cultivars of *V. oxycoccos* were selected for investigation. Fungi were isolated via the incubation of twigs, leaves and fruit on a PDA medium and identified according to their cultural and morphological characteristics. Microscopic fungi belonging to 14 genera were isolated from cranberry leaves and twigs, with *Physalospora vaccinii*, *Fusarium* spp., *Mycosphaerella nigromaculans* and *Monilinia oxycocci* being the most frequently isolated fungi. ‘Vaiva’ and ‘Žuvinta’ cultivars were the most susceptible to pathogenic fungi during the growing season. Among the clones, 95–A–07 was the most susceptible to *Phys. vaccinii*, 95–A–08 to *M. nigromaculans*, 99–Ž–05 to *Fusarium* spp. and 95–A–03 to *M. oxycocci*. Microscopic fungi belonging to 12 genera were isolated from cranberry berries. The most prevalent pathogenic fungi *M. oxycocci* were isolated from the berries of the cultivars ‘Vaiva’ and ‘Žuvinta’ and clones 95–A–03 and 96–K–05.

## 1. Introduction

The European cranberry (*Vaccinium oxycoccos* L., syn. *Oxycoccos palustris* Pers.) produces berries, which are an excellent source of bioactive, especially polyphenolic, compounds (i.e., flavonoids, anthocyanins and phenolic acids). Therefore, they are very beneficial for consumers’ health, and the consumption of these fruits and their products is widely recommended [1,2,3]. 

The geographical distribution of European cranberry is wide. *V. oxycoccos* is commonly known as “small cranberry” and found in native populations in forest areas in Europe, Asia and North America [4]. European cranberry is a common species in the oligotrophic and mesotrophic bogs. In Lithuania, the European cranberry is common; it grows in uplands and intermediate-type bogs, and less often in lowland bogs and waterlogged forests. The natural resources of the European cranberry are disappearing for various reasons. One of the most important of them is the effects of land reclamation. Aiming to preserve genetic resources, a unique collection of cranberry clones has been established in Vytautas Magnus University Botanical Garden. This collection promotes interest in the cultivation and breeding of European cranberry and becomes the basis for the preservation of selected valuable clones and for investigations of their intraspecies diversity, biochemical properties and bioactive compounds [5]. Individual cranberry clones of the same population can differ greatly, not only in morphological characteristics, but also in resistance to diseases [6]. The cultivars of other species of American cranberry (*Vaccinium macrocarpon* Aiton, syn. *Oxycoccus macrocarpus* (Aiton) Purs.) are known for their exceptional economical characteristics. The European cranberry is less economically valuable in comparison with the American cranberry; however, new productive cultivars bred in Estonia and Lithuania have economic potential, especially taken into account that the consumption of these valuable berries is a long-lasting tradition [7].

Fungi inhabiting the *Vaccinium* species play an important role in plants’ growth and can have a negative impact on the quality of yield [8]. Cranberries can be a potential reservoir of fungal pathogens [3]. Fungal diseases, particularly cranberry fruit rot, have been a serious problem resulting in the limiting of fruit production [9]. The mycological situation of the American cranberry (*V. macrocarpon*) has been studied extensively, but there are only a few studies focused on the European cranberry (*V. oxycoccos*) [3]. The diversity of fungal species isolated from the American cranberry may reach more than 60 species, including saprotrophic and pathogenic fungi, using a latent infection mode. It is well established that cranberry fruit rot is a complex disease caused by over 15 different species of fungi [10]. In Lithuania, *Phomopsis vaccinii*, the most harmful pathogen of *V. macrocarpon* were identified for the first time in 2002 [11]. However, other fungi, which can cause significant damage to the European cranberry during the growing season and reduce their yields, remain unstudied. 

To promote the breeding and cultivation of cranberries, it is necessary to investigate the characteristics of cranberries that are important for horticulture. It is important to choose natural clones of cranberries which have the highest productivity and disease resistance. Mycological studies of the European cranberry will contribute to the justification of the prospects for the cultivation of this species and to the clarification of the diversity of pathogenic fungi on different clones and cultivars. To gain better knowledge of the pathogenic fungi associated with *V. oxycoccos*, the objectives of this study were to detect the causal agents, which could cause *V. oxycoccos* diseases in Lithuania. This could be one of the strategies to control diseases caused by various pathogens. The aim of this study was to describe the diversity of fungi of *V. oxycoccos* clones and cultivars growing in Lithuania.

## 2. Results

### 2.1. Diversity of Fungi Associated with European Cranberry Twigs and Leaves

In total, 447 fungal isolates were obtained from cranberry twigs and leaves. These isolates represented 14 different fungal genera (Table 1). Fungi grew out of all samples collected (*n* = 68) from 17 European cranberry clones and yielded 297 isolates. A total of 150 fungal isolates were obtained from 5 cranberry cultivars with 20 collected samples (*n* = 20). The total amount of fungal isolates of cranberry clones and cultivars is presented in Figure 1. 

*Phys. vaccinii, Fusarium* spp., *M. nigromaculans* and *M. oxycocci* were the most frequently occurring fungi isolated from cranberry twigs and leaves. The number of isolates of *Alternaria* spp. and *B. cinerea* varied from 4% to 8%. Other fungi varied from 0.7% to 4% in cranberry twigs and leaves. 

The most common species from clones and cultivars was *Phys. vaccinii* (75 and 52 isolates, respectively), where relative abundance was 25.3% and 34.7%, respectively (Table 1). The species with the lowest number of isolates from cranberry clones were *Colletotrichum* spp. and *Trichoderma* spp., with nine isolates each and a relative abundance of 3%. The species with the lowest numbers from cultivars were *Colletotrichum* spp. and *Exobasidium vaccinii*, with four isolates each and a relative abundance of 2.7%.

The highest number of fungi (>40%) was detected from 27.3% of the examined plants (Figure 1). The highest occurrence of fungi was accounted for in the plants of the cranberry cultivars ‘Vaiva’, ‘Žuvinta’, ‘Amalva’ and ‘Vita’ and clones 96–Ž–13, 95–A–03. The lowest occurrence of fungi (<7%) was established in 9.1% of the examined plants. The occurrence of fungi in clones 97–J–07 and 99–Ž–11 is significantly lower compared to other clones and cultivars.

*Phys. vaccinii* was detected in 26.8% of cranberry plants (Figure 2a). The results showed the occurrence of the fungi in seven clones and three cultivars of cranberry, which varied from 7.5% to 16.3%. *Phys. vaccinii* was most abundantly found in cranberry clones 95–A–07, 95–A–03 and 96–K–10 (16.3%, 12.5% and 12.5% of plants infected, respectively), which accounted for 30% of the plants with the highest occurrence of fungi. *Phys. vaccinii* was isolated in the plants of three cultivars, and the highest occurrence (10%) was found in the cultivar ‘Vaiva’. The occurrence of these fungi in ‘Vita’ and ‘Amalva’ cultivars was 8.8%. However, no significant differences were established among individual clones and cultivars in relation to the occurrence of *Phys. vaccinii*. 

*Fusarium* spp. fungi were identified in 20.5% of the tested plants (Figure 2b). The occurrence of these fungi in six clones and three cultivars was low, varying from 7.5% to 25%. *Fusarium* spp. were most abundantly isolated from clone 99–Ž–05, however no significant difference in the occurrence of *Fusarium* spp. was established compared to other clones and cultivars. 

*M. nigromaculans* was not common in cranberries, with it being isolated from 8.6% of all tested plants (Figure 2c). Among the 17 clones investigated, it was found in 4 clones, with the highest occurrences—25% and 22.5%—in clones 95–A–08 and 96–Ž–13, respectively. *M. nigromaculans* was found only in plants of the ‘Amalva’ cultivar. Due to the large variation in the data, no significant differences were detected among clones and cultivars.

*M. oxycocci* occurred in 6.1% of all tested plants (Figure 3). The occurrence of fungi (>30%) was determined in 22.2% of plants. *M. oxycocci* was the predominant fungal species detected across the five clones and four cultivars. In cranberry clones 95–A–03 and 97–J–04 the occurrence of the pathogen was 32.5% and 27.5%, respectively. The most sensitive cultivars to these fungi were ‘Žuvinta’ and ‘Vaiva’, with an occurrence of 36.3% and 26.3%, respectively, for *M. oxycocc*. Plants of clone 99–Ž–10 were significantly more infected by *M. oxycocci* compared to other clones and cultivars.

The clones and cultivars more frequently infected by pathogens were identified in this study. The highest level of *Phys. vaccinii* infection was revealed in cranberries of clones 95–A–07 and 95–A–03. Clones 95–A–03 and 97–J–04 were the most sensitive to *M. oxycoccii* (32.5% and 27.5%, respectively), 95–A–08 and 96–Ž–13 to *M. nigromaculans* (25% and 22.5%, respectively) and 99–Ž–05 to *Fusarium* spp. (25%).

The cultivar ‘Amalva’ was one of the most sensitive cultivars to fungal pathogens. Leaves and other parts of this cultivar had higher occurrence of *Phys. vaccinii* (8.8%) and *M. nigromaculans* (17.5%) compared to other cultivars. The sensitivity of the cultivar ‘Vaiva’ to *Phys. vaccinii* (10%) and *M. oxycocci* (26.3%) was also evident in this study. *Phys. vaccinii* was prevalent in 8.8% of the plants of the cultivar ‘Vita’, whereas *Fusarium* was found in 10% of the plants. *M. oxycocci* and *Fusarium* were the most abundant in the leaves and twigs of cultivar ‘Žuvinta’, with, respectively, 32.5% and 10% occurence. The lowest number of pathogens was determined in cranberries of the cultivar ‘Reda’. Only a 10% occurrence of *Fusarium* spp. was determined in the ‘Reda’ cultivar.

### 2.2. Diversity of Fungi Associated with European Cranberry Berries

A total of 468 isolates from 12 genera were isolated from 22 samples of cranberry berries (Table 2). It was determined that 16% of the berries collected for the study had typical damage symptoms of rot. 

The most common species from clones and cultivars was *M. oxycocci* (79 and 26 isolates, respectively), and the relative abundance was 22.3% for clones and 22.8% for cultivars (Table 2). The species with the lowest numbers of isolates from clones and cultivars was *Discosia* sp. (eight and two isolates, respectively), and the relative abundance was 2.3% for clones 1.8% for cultivars.

The highest occurrence of fungi was isolated from the berries of cultivars ‘Vita’ (20%), ‘Amalva’, ‘Žuvinta’ (17.5% each) and ‘Vaiva’ (13.8%) (Figure 4). Among the clones, the highest occurrence of fungi (>20%) was determined in the berries of 95–A–03, 99–Ž–06 and 99–Ž–11. The berries of the ‘Reda’ cultivar were significantly less infected by fungi.

*M. oxycocci* occurred in 22.4% of all tested berries. It was found in the berries of six clones and three cultivars (Figure 5). *M. oxycocci* particularly occurred in the berries of ‘Vaiva’ and ‘Žuvinta’. *M. oxycocci* was also the most common in the plants of these cultivars during study. Among the clones, the highest occurrence of this fungus was found in the berries of clones 95–A–03 and 96–K–05.

In this study, a different occurrence of *M. oxycocci* was found in plants and berries. The high occurrence (7.5%) of *M. oxycocci* was isolated in the berries of clone 99–Ž–11, but during the study, these fungi were not common on the plants; only 1.2% occurrence was determined. In the berries of clones 95–A–03 and 97–J–04, the occurrence of *M. oxycocci* was not high, reaching, 10% and 8.8%; however, during the study, the fungus was isolated the most abundantly in the plants of these clones, reaching 32.5% and 27.5%. The positive (*r*^2^ = 0.52, *p* < 0.01) correlation was found between *M. oxycocci* on twigs and leaves of cranberry and rotten cranberry berries.

### 2.3. The Abundance and Ecological Parameters of Fungi Associated with European Cranberry

The most common species on *V. oxycoccos* twigs and leaves was *Phys. vaccinii* (127 isolates), where the relative abundance was 28.4%, and *M. oxycocci* (105 isolates) on berries, where the relative abundance was 22.4% (Table 3). 

The species with the lowest numbers of isolates on twigs and leaves was *Colletotrichum* spp., with the number of isolates being 13 with a relative abundance of 2.9%. The species with the lowest numbers in berries was *Discosia* sp. (10 isolates), with a relative abundance of 2.2%. 

The species richness on cranberry twigs and leaves was higher than on berries (Table 3). The Shannon–Wiener diversity index indicated that the species diversity on cranberry twigs, leaves and berries was low. Based on the Simpson’s dominance index value, it was found that the structure of fungal communities on twigs, leaves and in berries was stable. The dominance index on cranberry twigs and leaves is higher than on berries, because twigs and leaves were dominated by *Phys. vaccinii*, showing a relative abundance of 28.4%.

The positive correlation (*r*^2^ = 0.66, *p* < 0.01) between the total count of pathogenic fungi on twigs and leaves and the total count of pathogenic fungi of cranberry berries was determined. 

The species richness index on cranberry twigs and leaves from clones was higher than from cultivars (Table 4). The Shannon–Wiener diversity index indicated that the species diversity from cranberry clones and cultivars was low. Based on the Simpson’s dominance index value, it was found that the structure of fungal communities from both clones and cultivars was stable. The dominance index from cranberry cultivars was higher than from clones, because from cultivars it was dominated by *Phys. Vaccinia*, showing a relative abundance of 34.7%. 

The species richness in cranberry berries from clones was higher than from cultivars. The Shannon–Wiener diversity index, which considers relative abundance between species, species richness and evenness, indicated that the species diversity from clones and cultivars was low. The Simpson’s index from clones and cultivars stated that the fungal community structure was stable. The dominance index from cultivars was slightly higher than from clones because *M. oxycocci* dominated with a relative abundance of 22.8%. The similarity index between clones and cultivars was 3.9. If the similarity index between clones and cultivars was lower than 0.50, it could be concluded that fungal communities of clones and cultivars were the same.

The positive correlation (*r*^2^ = 0.58, *p* < 0.01) between the total count of pathogenic fungi from clones on twigs and leaves and the total count of pathogenic fungi of cranberry berries from clones was determined. 

## 3. Discussion

In this study, microscopic fungi belonging to fourteen genera were isolated from the twigs and leaves of small cranberries (Table 2). These fungi are common in plants of the *Ericaceae* family [12]. Many of them were represented by fungi that were not pathogenic to cranberry plants and generally considered to be saprotrophic, such as *Alternaria* and *Cladosporium* spp., which are typical airborne fungi. These fungi were less frequently isolated in this study. The genus *Botrytis*, which causes field and storage problems [13], was recorded in twig and berry samples. The most common and threatening fungi in Central Europe are *Verticillium* and *Phytopthora*, as well as *B. cinerea* and *Colletotrichum acutatum*, which are involved in yield and quality losses of berry fruits [14]. In this study, *Phytophthora*, *Trichoderma* and *Verticillium* were rarely isolated in cranberry samples because they spread in the rhizosphere, and it is possible for them to transfer onto the twigs from there. *Phytophthora* fungi can be found on cranberries growing in low bogs [15] and might pose a threat to natural ecosystems [16]. *Trichoderma* spp. is a fungus found in many types of environments [17]. *Verticillium* spp. are opportunistic fungi that persist in the soil as saprophytes [14].

According to the results of the study, the most frequently isolated fungi were *Phys. vaccini, Fusarium* spp., *M. nigromaculans* and *M. oxycocci,* which can significantly affect cranberry yields. These fungi are known as cranberry twigs, leaf and fruit rot pathogens [12]. For comparison, *Botrytis cinera*, *Fusiccocum putrefaciens*, *Phomopsis vaccinii*, *Pestalotia vaccinii*, *Discosia artocrea* and *Phys. vaccinii* prevailed in *Vaccinium macrocarpon* Ait. during the growing season in Latvia [12]. In Poland, according to Michalecka et al. [18], a significant proportion of the fungi on the twigs and leaves of the American cranberry were *Diaporthe vaccinii*, *Ramularia* spp. and *Erysiphe* spp. These pathogens reproduced on plants and berries during the growing season and, if they survive, increased the disease incidence of cranberries in the following year. This suggests that most of the fungi associated with cranberry are generally able to thrive in a wide variety of environmental conditions and can use a variety of different resources during the growing season [19]. It is generally assumed that fungal diversity and colonization levels increase during the season and host tissue aging [20].

Even though *M. oxycoccos* is the causal agent of cranberry mummies, it was not particularly common in our study. The fungus is widespread and has been recorded in America and Europe (Germany, Denmark and Italy) [21]. Under Lithuanian conditions, it is one of the most damaging pathogens of cranberries, leading to wilting, the toppling of the plant and the mummification of berries [5]. Pathogens of this genus are also widespread in American cranberry plantations. However, according to Conti et al. [22], *M. oxycocci* has been found in American cranberries only in organic farms, indicating that fungicides have a strong and persistent effect on the fungus population. 

*M. nigromaculans* is common in all cranberry sites and often associated with red leaf spots and the blackening of stems [23]. The fungus forms perithecia in early autumn. *V. oxycoccos* is recorded as an intermediate plant and *V. macrocarpon* as a host plant [24]. 

*Phys. vaccini* was found on cranberries in our study. *Phys. vaccini* causes serious diseases of cranberries, evident in upright dieback and blotch rot [25]. The fungus persists as a latent infection in leaves during the growing season, forming fruiting bodies as the leaves die. *Phys. vaccini* is very common in American cranberries, damaging berries not only during the growing season but also during storage, causing blotch rot. *Phys. vaccini* can live as an endophyte and as a nectrotroph [19]. In our study *Phys. vaccini* was frequently detected in European cranberries. Similar results were obtained in American cranberries in Latvia [12]. Tadych et al. [26] suggest that the prevalence of this pathogen is closely related to climatic conditions, Olatinwo et al. [10] associated it with a place of cultivation and McManus et al. [27] associated it with different phenological stages of the plant.

*Fusarium* spp. fungi cause more damage in old growing areas [28]. *Cladosporium* spp., *Alternaria* spp. and *Penicillium* spp. were isolated from cranberry twigs and leaves, with pathological relationships of minor importance on cranberry leaves; however, relatively early in the growing season, the spread of *Alternaria* spp. and *Penicillium* spp. can cause fruit spoilage during storage [23,29]. *Alternaria* and *Fusarium* can cause fruit, leaf, stem or root diseases in cranberry, not only directly but also as secondary pathogens [30]. In our study, the fungi *Alternaria* spp. and *Cladosporium* spp. were mostly found on the surface of the cranberry twigs. *Colletotrichum* spp. was found to be widespread in large cranberry sites in Poland and America [10,18]. These fungi can overwinter in the plant’s circulatory tissues and fallen leaves and infect newly formed shoots in spring [27]. Less common fungi *B*. *cinerea* caused upright dieback, blossom and ovaries blight, and yellow rot in Latvia [12]. 

The productivity of the European cranberry in its natural habitat depends on a variety of factors, including ecological factors [31]. When assessing the productivity of individual cranberry cultivars and clones in a botanical garden, it was found that yields vary considerably because of meteorological conditions during berry growth and ripening [5]. However, pathogenic fungi are another important factor that can affect cranberry yields. It has been suggested that differences in the incidence of field cranberry fruit rot among cultivars may reflect host plant resistance [32]. Fungi causing fruit rot in cranberry fields can reduce yield by up to 33% [33] or even 50% [28]. Fruit rot is an important problem in all plantations of cranberry [34]. Cranberry fruit rot fungi exist as indigenous populations in cranberry growth sites [10]. Doyle et al. [28] reported that 12 fungal complexes can damage berries. In our study, fungi belonging to 12 genera were isolated. *M. oxycocci*, *Cladosporium* spp., *Alternaria* spp., *Penicillium* spp., *Colletotrichum* spp. and *Phys. vaccinii* found in rotten berries indicate that the berries are infected during the growing season. *Penicillium* spp. causing damage to stored products were also isolated during the growing season. Michalecka et al. [18] suggest that cranberry fruits are infected at all stages of the growing season, but infections remain latent until fruit maturity. Many fungal pathogens may be latent in their hosts long before the outbreak of disease symptoms. In a study of Conti et al. [22], four species were consistently prevalent in American cranberry before harvest: *Godronia cassandrae*, *Colletotrichum fructivorum*, *Allantomopsis cytisporea* and *Coleophoma empetri*. Oudemans et al. [32] found *Coleophoma empetri, Colletotrichum empetri, M. oxycocci* and *Phys. vaccinii.*

In our study, *M. oxycocci* was one of the most frequently isolated pathogens in cranberry berries. The fungal infection was the most abundant in the berries of the cultivars ‘Žuvinta’ and ‘Vaiva’, and *M. oxycocci* was also predominant in the twigs and leaves of these cultivars. Oudemans et al. [32] stated that the cranberry fruit rot fungi infect the host early in the growing season, during bloom, and then remain latent until ripening.

In small cranberries, *Phys. vaccinii* is rarely found, but according to Conti et al. [22], it was one of the most frequently detected pathogens in American cranberries. Olatinwo et al. [10] reported that this pathogen is more common in ripe large cranberries before harvest, and less common in rotten berries. According to McManus [27], *Phys. vaccinii* can also be isolated from healthy berries (10–30%), indicating that fungal infection does not always cause disease.

*Colletotrichum* spp. have been rarely found on plants studied but are an important pathogen of cranberries that can cause the rotting of both ripening and stored fruit [3]. C. *acutatum* is mentioned as one of the most common pathogens on ripe berries in pre-harvest studies by Conti et al. [22].

Other fungi isolated from the aerial parts of cranberries, such as *Cladosporium* or *Alternaria*, are not significant causal agents of fruit rot [35]. These fungi are mentioned by Olatinwo et al. [10] as common saprotrophs on cranberry berries, which overwinter and sporulate on various other host plants before infecting cranberries. In our study, *Alternaria* spp. were abundantly isolated from the leaves and twigs of the plants, and *A. alternata* was also abundantly found in the berries. Studies have shown that *A. alternata* can dominate on plants and berries in the lethal phase until harvest [36].

In this research, it became clear that cranberry berries ripening during mid-ripening season (from 11 to 20 September) were more abundantly infected by fungi than those ripening later (from 22 to 30 September). Fungal occurrence is likely determined by genetic interactions of fungi with cranberry during both primary and secondary infections stages, temporal overlap of conidia production and bloom and environmental influences such as temperature and moisture [27]. 

The study revealed that fungi predominated in the cultivars ‘Vaiva’ and ‘Žuvinta’ and clones (95–A–03, 95–A–07, 95–A–08, 96–Ž–13 and 99–Ž–05), causing twig and leaf diseases and berry rot (clones 95–A–03 and 96–K–05). The differences in susceptibility of the cultivars are difficult to assess and they may be related to the particularities of the pathogen’s spread and dependence on environmental conditions. Different fungal species predominate at different locations, and the frequencies can vary greatly over years [10]. Cicalese et al. [35] were unable to detect any differences in resistance between the American cranberry cultivars tested. Few differences between rot-resistant and rot-susceptible cultivars were found, but most samples collected at the same sampling time point, regardless of rot resistance or susceptibility, showed a high level of similarity in fungal diversity and population composition [22]. 

Many bioactive compounds of cranberry can function as constitutive or inducible barriers against microbial pathogens, and bioactive compound composition can change in response to microbial attack [37,38]. Tadych et al. [26] found that benzoic acid and quinic acid reduced the growth of four tested cranberry rot fungi (*Coleophoma empetri*, *Colletotrichum gloeosporioides*, *Phyll. vaccinii* and *Phys. vaccinii*). To reduce plant disease, it may be a viable strategy to select crop plants that maintain higher levels of organic acids or other potential virulence suppressors through plant development. 

## 4. Materials and Methods

### 4.1. Sampling Collection

Twigs, leaves and berries of *V. oxycoccos* were collected in 2021 in the field collection of Vytautas Magnus University Botanical Garden, which represents the unique collections of genetic resources of European cranberry in Central Lithuania. *V. oxycoccos* clones were collected between 1998–1999 from Čepkeliai, Kamanos and Žuvintas reserves. Afterward, the clones were multiplied and planted in the outdoor collection of Vytautas Magnus University Kaunas Botanical Garden. Cranberries have been cultivated under uniform ecological conditions in acid peat beds, pH 3.5–5.0. Seventeen clones and five cultivars of *V. oxycoccos* were selected for investigation (Table 5).

The growing season of cranberry lasts for approximately 180–190 days, with an average temperature of −6 °C in January and +16 °C in July. The average annual precipitation during the growing season is 600 mm.

In total, 88 samples of cranberry twigs with leaves were analyzed. They were collected in the period from the last week of May to the third week of June and when the berries were ripe, before harvesting in the third week of September. Four samples of cranberry twigs with leaves were randomly taken from each sampling plot (1 m^−2^). The sample consisted of 10 pieces (approximately 5 cm long) of twigs with leaves. The samples were stored at +4 °C for 1 to 4 days until further examination. 

Samples of berries were collected before the harvest (in the third to fourth weeks of September). Approximately 50 g of berry samples were collected from a single genotype in each sampling plot. Rotten berries (e.g., berries showing CFR symptoms, as described by Oudemans et al. [32] and Polashock et al. [36]) were sampled. In total, 22 samples of berries were collected. The samples were stored at +4 °C until the analysis. 

### 4.2. Isolation and Identification of Fungi

Twigs of cranberry were cut into short (~1 cm) pieces with a sterile scalpel. Twigs and leaves were washed in distilled water, placed for 1 min in 2% sodium hypochlorite (NaOCl) solution, then washed again in sterile distilled water 3 times (3 min each time). The surface-disinfected pieces were dried completely and cultured on a PDA medium in Petri dishes and incubated at 25 °C under photoperiod 12/12 h (light/dark) conditions for 14–21 days. 

The surfaces of berries have been disinfected by brief immersion in 95% ethanol followed by 5 min in 0.5% NaOCl. After briefly rinsing berries in sterile distilled water three times, calyces were removed. Ten pieces of randomly selected (including both rotten and healthy) berries of each clone and cultivar were cut longitudinally and placed onto Potato-Dextrose agar (PDA, Sigma-Aldrich) amended with streptomycin sulfate (100 µL mL^−1^) to inhibit bacterial growth. For each berry, one half was placed with its cut surface down onto a PDA medium. Each 90-mm Petri dish contained three berry halves. Berries were incubated at 25 °C, in the dark, for 10–14 days. 

The colonies of each fungal morphotype were grown into pure cultures using the three-point inoculation method on CDA (Czapek Dox Agar media; Biolife, Italy) and PDA mediums and incubated at 25 °C for 10–14 days or 15–30 days for slowly growing species. Fungal identification was based on cultural (colony size and pattern, color in colony surface and reverse, colony surface texture) and morphological characteristics (type, size and shape of conidia, fruit bodies and others). Slower growing fungi and fungi which did not produce spores were transferred to new plates. The fungal strains which failed to sporulate following incubation (for 30 days) were considered *Mycelia sterilia*. Yeasts were distinguished from filamentous fungi by colony and cell morphologies. The fungal isolates were identified using various identification keys [32,39,40,41,42].

The frequency of occurrence (FO) of a fungal species was determined as the percentage of samples in which the species was found compared to the total number of samples used in the study. 

### 4.3. Statistical Analysis

The statistical analysis of the study data was performed using the software One-way ANOVA from the statistical analysis package STATISTICA version 10. Duncan’s criterion and the LSD test were used to assess the significance of the differences between clones and cultivars [43]. The research data that did not conform to the normal distribution were transformed using the mathematical function *y* = *log*(*x* + 1) prior to statistical evaluation. The differences between averages of clones and cultivars, marked by different letters, are significant at a 95% probability level (*p* < 0.05). Significant interaction between individual values obtained using the Pearson correlation coefficient.

The data were analyzed for the relative abundance and similarity index [44], species richness [45], Shannon–Wiener diversity index [46] and Simpson’s dominance index [47].

## 5. Conclusions

This study evaluated the diversity of microscopic fungi in 17 clones and 5 cultivars of European cranberry plants. Microscopic fungi belonging to 14 genera were isolated from cranberry leaves and twigs, and 12 genera were isolated from berries.

The results showed that *Phys. vaccinii*, *Fusarium* spp., *M. nigromaculans* and *M. oxycocci* were the dominant pathogenic fungi on cranberry twigs and leaves. *Phys. vaccinii* was the most abundantly isolated in cultivar ‘Vaiva’; *Fusarium* spp. in cultivars ‘Reda’, ‘Vita’ and ‘Žuvinta’; *M. nigromaculans* in cultivar ‘Amalva’ and *M. oxycocci* in cultivars ‘Žuvinta’ and ‘Vaiva’. Among the clones, *Phys. vaccinii* was the most abundant in clone 95–A–07, *Fusarium* spp. in clone 99–Ž–05, *M. nigromaculans* in clone 95–A–08 and *M. oxycocci* in clones 96–Ž–13 and 95–A–03.

*M. oxycocci* was the most abundant in rot-damaged berries. ‘Vaiva’ and ‘Žuvinta’ cultivars berries were established with the highest *M. oxycocci* occurrence. Among the clones, the highest occurrence of *M. oxycocci* was established on the 95–A–03 and 96–K–05 clone berries.

The species richness on cranberry twigs and leaves was higher than on berries. The species diversity on cranberry twigs, leaves and berries was low. The structure of fungal communities on twigs, leaves and berries was stable.

## Figures and Tables

**Figure 1 plants-12-02360-f001:**
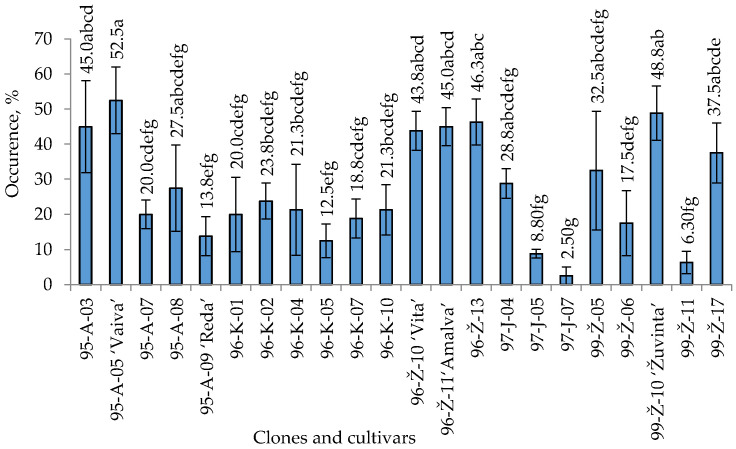
The fungal occurrence on *V. oxycoccos* twigs and leaves. Note: The differences between the averages of clones and cultivars marked by different letters are significant (*p* < 0.05). Whiskers indicate standard error of the mean.

**Figure 2 plants-12-02360-f002:**
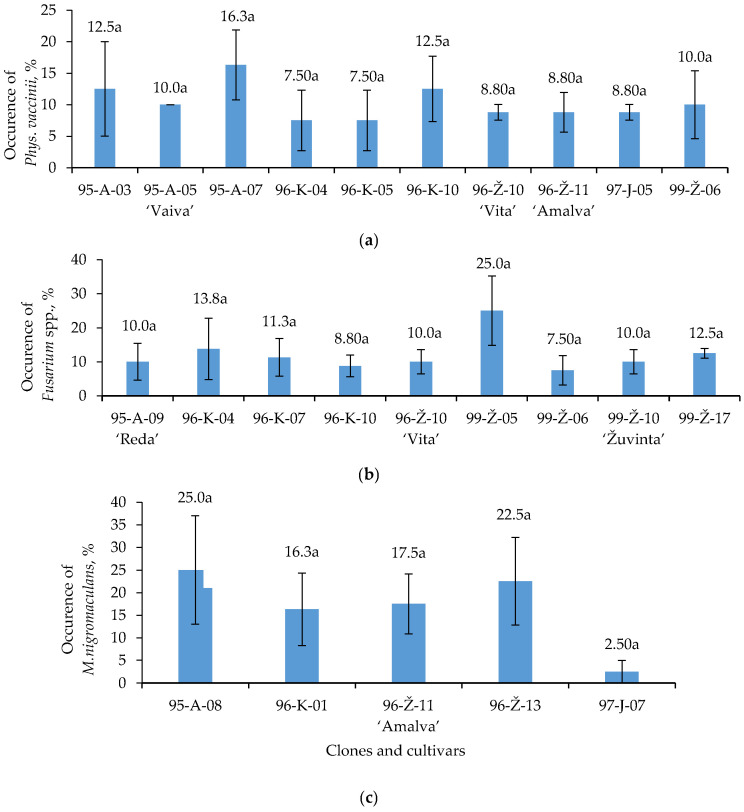
*Phys. vaccinii* (**a**), *Fusarium* spp. (**b**) and *M. nigromaculans* (**c**) occurrence on *V. oxycoccos* twigs and leaves. Note: There is no significant difference (*p* > 0.05). Whiskers indicate standard error of the mean (*n* = 4).

**Figure 3 plants-12-02360-f003:**
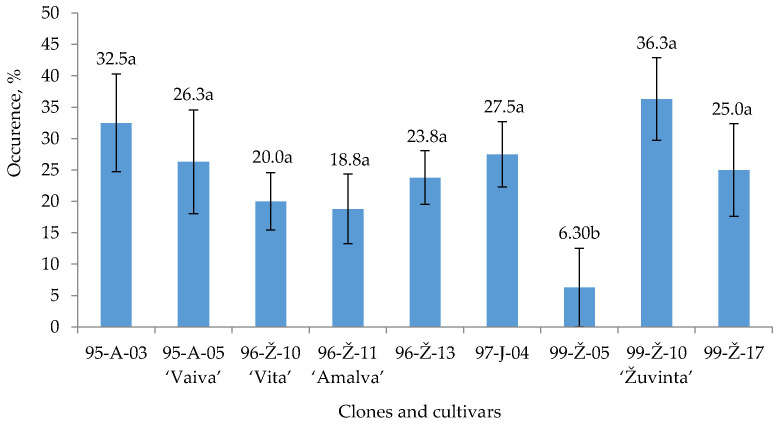
*M. oxycocci* occurrence on *V. oxycoccos* twigs and leaves. Note: The differences between the averages of clones and cultivars, not marked by the same letter, are significant (*p* < 0.05). Whiskers indicate standard error of the mean (*n* = 4).

**Figure 4 plants-12-02360-f004:**
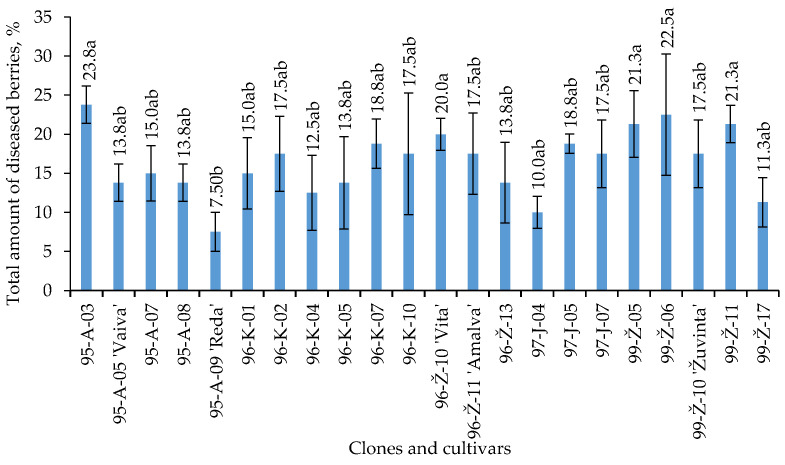
The total amount of fungi occurrence in V. oxycoccos berries. Note: The differences between the averages of treatments, not marked by the same letter (a,b), are significant (*p* < 0.05). Whiskers indicate standard error of the mean.

**Figure 5 plants-12-02360-f005:**
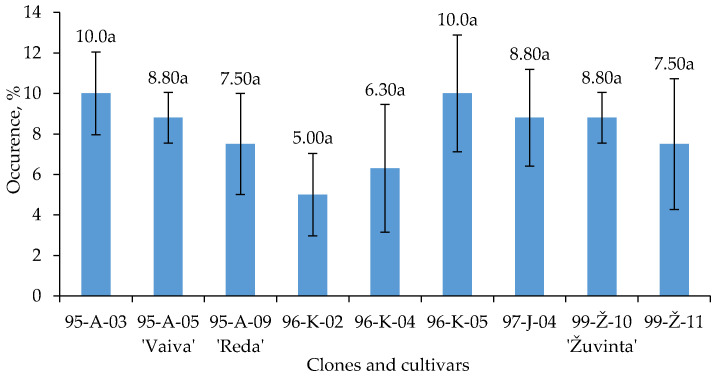
*M. oxycocci* occurrence in *V. oxycoccos* berries. Note. There is no significant difference (*p* > 0.05). Whiskers indicate standard error of the mean (*n* = 4).

**Table 1 plants-12-02360-t001:** Fungi associated with European cranberry twigs and leaves.

Fungal Species	Pathogenicity *	No. of Isolates (%)	
		Clones	Cultivars
*Alternaria* spp.	–	15 (5.1)	12 (8)
*Botrytis cinerea*	+	13 (4.4)	6 (4)
*Colletotrichum* spp.	+	9 (3)	4 (2.7)
*Cladosporium* spp.	–	12 (4)	5 (3.3)
*Exobasidium vaccinii*	+	10 (3.4)	4 (2.7)
*Fusarium* spp.	+	68 (22.9)	29 (19.3)
*Mycosphaerella nigromaculans*	+	34 (11.4)	11 (7.3)
*Monilinia oxycocci*	+	21 (7.1)	8 (5.3)
*Phylosticta* spp.	+	11 (3.6)	6 (4)
*Pestalotia vaccinii*	+	10 (3.4)	5 (3.3)
*Physalospora vaccinii*	+	75 (25.3)	52 (34.7)
*Phytophthora* spp.	+	5 (1.7)	1 (0.7)
*Trichoderma* spp.	–	9 (3)	3 (2)
*Verticillium* spp.	–	5 (1.7)	4 (2.7)
Total		297 (100)	150 (100)

Note. * Plant pathogen “+”.

**Table 2 plants-12-02360-t002:** Fungi associated with European cranberry berries.

Fungal Species	Pathogenicity *	No. of Isolates (%)	
		Clones	Cultivars
*Alternaria* spp.	–	64 (18)	17 (14.9)
*Botrytis cinerea*	+	27 (7.6)	9 (7.9)
*Cladosporium oxycocci*	–	34 (9.5)	12 (10.5)
*Coleophoma empetri*	+	17 (4.8)	8 (7)
*Colletotrichum* spp.	+	29 (8.2)	10 (8.8)
*Discosia* sp.	+	8 (2.3)	2 (1.8)
*Fusarium* spp.	+	19 (5.3)	7 (6.1)
*Monilinia oxycocci*	+	79 (22.3)	26 (22.8)
*Penicillium* spp.	+	18 (5.1)	6 (5.3)
*Pestalotia* sp.	+	12 (3.4)	4 (3.5)
*Phyllosticta* sp.	+	9 (2.5)	3 (2.6)
*Physalospora vaccinii*	+	39 (11)	10 (8.8)
Total		355 (100)	114 (100)

Note. * Plant pathogen “+”.

**Table 3 plants-12-02360-t003:** Relative abundance and ecological parameters of fungi associated with the European cranberry.

Fungal Species	Relative Abundance of Fungi (%)	
	Twigs and Leaves	Berries
*Alternaria* spp.	6	17.3
*B. cinerea*	4.3	7.7
*Cladosporium* spp.	3.8	9.8
*Coleophoma empetri*	-	5.3
*Colletotrichum* spp.	2.9	8.3
*Discosia* sp.	-	2.2
*E. vaccinii*	3.1	-
*Fusarium* spp.	21.7	5.5
*M. nigromaculans*	10.1	-
*M. oxycocci*	6.5	22.4
*Phyllosticta* spp.	3.8	2.6
*Penicillium* spp	-	5.1
*Pestalotia vaccinii*	3.4	3.4
*Phys. vaccinii*	28.4	10.4
*Phytophthora* spp.	1.3	-
*Trichoderma* spp.	2.7	-
*Verticillium* spp.	2	-
Total	100	100
Ecological parameters		
Species richness index	3.24	2.63
Shannon–Wiener diversity index	0.68	0.51
Simpson dominance index	0.51	0.42
Similarity index	5.2	

**Table 4 plants-12-02360-t004:** Ecological parameters of fungi associated with the European cranberry.

Ecological Parameters	Twigs and Leaves		Berries	
	Clones	Cultivars	Clones	Cultivars
Species richness index	2.98	2.86	2.94	2.79
Shannon–Wiener diversity index	0.71	0.60	0.66	0.61
Simpson dominance index	0.34	0.30	0.38	0.40
Similarity index	4.5		3.9	

**Table 5 plants-12-02360-t005:** List of *V. oxycoccos* clones and cultivars investigated in the study.

Clone	Ripening Season
95–A–03	Mid
95–A–07	Mid
95–A–08	Mid
96–K–01	Mid
96–K–02	Late
96–K–04	Mid
96–K–05	Mid
96–K–07	Mid
96–K–10	Late
96–Ž–13	Mid
97–J–04	Mid
97–J–05	Mid
97–J–07.	Mid
99–Ž–05	Mid
99–Ž–06	Mid
99–Ž–11	Late
99–Ž–17	Mid
Cultivars	
‘Amalva’ (96–Ž–11)	Mid
‘Reda’ (95–A–09)	Late
‘Vaiva’ (95–A–05)	Mid
‘Vita’ (96–Ž–10)	Late
‘Žuvinta’ (99–Ž–10)	Mid

Note: Mid-ripening from 11 to 20 September; late-ripening from 22 to 30 September.

## Data Availability

Not applicable.

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
