# Peer review of "Fungi Present in the Clones and Cultivars of European Cranberry (Vaccinium oxycoccos) Grown in Lithuania"

_plants, 2023, doi:10.3390/plants12122360_

Round 1
Reviewer 1 Report
The manuscript by Sinkevičienė and co-workers presents an enumeration of the fungal species present on European cramberry grown in a specific botanical garden in Lithuania.
Comments and questions
1. Line 39: ..."natural resources of the European cranberry are disappearing..." Not clear on the meaning of this: does it mean its only picked in the wild rather than cultivated??
Line 80: "A total of 150 80 fungal isolates were obtained from 5 cranberry cultivars with 20 tested samples (n=20)." What does the "20- tested samples" refer to?
Line 89, referring to Table 1: "Other fungi varied from 0.7% to 3.6% in 89 cranberry twigs and leaves" I see the 0.7 value, don't see the 3.6 and see other, various higher values.
Line 93: "species with lowest number of isolates from cranberry clones was Colletotrichum spp. with 93 9 isolates and a relative abundance of 3%." Tricoderma had the same abundance.
Fig. 1 legend, instead of "by not the same letter" can say "by different letters".
In Fig. 2, instead of "a", "b" and "c" labels, could put the entire name of the organism. For Fig. 2c, why is the y-axis compressed?
Line 141, "sensitive" refers to being more frequently infected by some fungus?
General comment in the results: it would be helpful to the reader if the Fig. or Table that's being referred to is cited more frequently- its easy to get lost if not.
Generally good, see my specific comments in Comments and Suggestions for Authors
Reviewer 2 Report
Overall a well written article on mushrooms related to the European cranberry (Vaccinium oxycoccos L.). They play an important role in plant growth and disease control. Authors The article presents the results of research on the diversity of fungi found on European cranberries grown in Lithuania, causing cranberry diseases. Seventeen clones and five cultivars of V. oxycoccos were tested. The fungi were isolated on PDA medium and identified on the basis of cultural and morphological features. Fungi belonging to 14 genera were isolated, most often Physalospora vaccinii, Fusarium spp., Mycosphaerella nigromaculans and Monilinia oxycocci. The cultivars 'Vaiva' and 'Žuvinta' were the most susceptible to pathogenic fungi during the growing season. The most common pathogenic fungus, M. oxycocci, was isolated from the fruits of the 'Vaiva' and 'Žuvinta' cultivars. Figures are legible with statistically significant differences marked. The tests used in the statistical analysis are correctly matched to the data. The References section could have been more extensive (broader literature review).
Round 2
Reviewer 1 Report
The authors have addressed the comments and concerns that I raised about the manuscript.
Essentially okay.